# Recent Progress on the Versatility of Virus-Like Particles

**DOI:** 10.3390/vaccines8010139

**Published:** 2020-03-20

**Authors:** Ciying Qian, Xinlin Liu, Qin Xu, Zhiping Wang, Jie Chen, Tingting Li, Qingbing Zheng, Hai Yu, Ying Gu, Shaowei Li, Ningshao Xia

**Affiliations:** 1National Institute of Diagnostics and Vaccine Development in Infectious Disease, School of Life Sciences, Xiamen University, Xiamen 361102, China; ciying@stu.xmu.edu.cn (C.Q.); huazhonglxl@163.com (X.L.); 21620191152724@stu.xmu.edu.cn (Q.X.); 21620190154550@stu.xmu.edu.cn (Z.W.); jiecc@stu.xmu.edu.cn (J.C.); litingmu@stu.xmu.edu.cn (T.L.); nsxia@xmu.edu.cn (N.X.); 2State Key Laboratory of Molecular Vaccinology and Molecular Diagnostics, School of Public Health, Xiamen University, Xiamen 361102, China; abing0811@xmu.edu.cn (Q.Z.); yuhai@xmu.edu.cn (H.Y.)

**Keywords:** virus-like particles, nanocarrier, recombinant vaccine

## Abstract

Virus-like particles (VLPs) are multimeric nanostructures composed of one or more structural proteins of a virus in the absence of genetic material. Having similar morphology to natural viruses but lacking any pathogenicity or infectivity, VLPs have gradually become a safe substitute for inactivated or attenuated vaccines. VLPs can achieve tissue-specific targeting and complete and effective cell penetration. With highly ordered epitope repeats, VLPs have excellent immunogenicity and can induce strong cellular and humoral immune responses. In addition, as a type of nanocarrier, VLPs can be used to display antigenic epitopes or deliver small molecules. VLPs have thus become powerful tools for vaccinology and biomedical research. This review highlights the versatility of VLPs in antigen presentation, drug delivery, and vaccine technology.

## 1. Introduction

Vaccination is one of the most cost-effective ways to control and prevent the spread of infectious disease [1]. Pathogens and their metabolites are prepared as an immunological formulation that can be used to prevent or treat infectious diseases by artificial attenuation, detoxification, inactivation, or genetic engineering methods [2]. Traditional vaccines, including live attenuated vaccines and inactivated vaccines, can induce strong and long-term effective immune responses in host cells, and have thus held a prominent position within the vaccine market. Although many traditional vaccines are safe and effective in humans, several inherent risks are associated with certain live attenuated and inactivated vaccines, including incomplete attenuation, unstable toxicity, and risk of mutation, which make it difficult to guarantee the longevity of these formulations [3,4]. Indeed, novel vaccines, such as genetically engineered vaccines and synthetic peptide vaccines, have received widespread attention, offering greater immunogenicity and safety, and this is particularly noted for genetically engineered vaccines based on virus-like particles (VLPs).

VLPs are non-genetic multimeric nanoparticles formed through the self-assembly of one or more viral structural proteins in vivo or in vitro. Having no viral genetic material, VLPs therefore have no pathogenicity or infectivity [5]. In addition, because of their similar conformation to that of natural viral particles, VLPs can mimic the antigenic epitopes of the natural virus [6]. Indeed, nanometer-sized VLPs comprising highly ordered repeat epitopes are ideal for cellular phagocytosis and antigen presentation of dendritic cells (DCs), thereby effectively inducing a strong cellular and humoral immune response [7]. Therefore, VLPs are expected to be ideal candidates for vaccine development, with both high immunogenicity and safety. They can also act as carriers of foreign antigens or adjuvants to stimulate the immune system. Here, we review the utility of VLPs in drug delivery and antigen presentation, and highlight the recent advances in VLP-based vaccine research (Figure 1).

## 2. VLP Technology Applied in Basic Research

Free of genetic material inside, VLPs have broad potential applications and are a hot topic in many current research areas. In terms of structure architecture, VLPs can be divided into two types: non-enveloped and enveloped (Figure 2). Non-enveloped VLPs, such as human papillomavirus (HPV) VLPs, are relatively simple in structure and consist of a single or multiple structural protein(s) of a specific virus [8,9]. Enveloped VLPs, such as influenza VLPs, have a complex structure constituted by both virus and host membrane components [10,11]. In addition to their capability to stimulate an immune response, both forms of VLPs can be manipulated into a chimeric structure to display foreign antigens for a broader immunity, and can be repurposed as highly specific nanocarriers for the delivery of antigens or therapeutic molecules.

### 2.1. VLP Immunogens

VLPs have become important tools in the biomedical field. VLPs are capable of self-assembly, they have a repetitive order and, because of the absence of a viral genome, VLPs can confer high immunogenicity and antigenicity [5]. VLPs are commonly 20~200 nm in diameter [12], which is close to the optimal diameter recognized by DCs and antigen-presenting cells (APCs). As such, VLPs are easily absorbed by APCs and transported to lymph nodes. This, in turn, stimulates the secretion of various cytokines by APCs to induce a strong immune response [13]. Furthermore, as a regular polyhedron with a highly repetitive epitope, VLPs can cross-link B-cell receptors and effectively activate B-cells [14,15]. In the absence of DCs, B cells are sufficient to induce T follicular helper cell development [16]. Moreover, activated B cells can promote CD4+ T cell proliferation and differentiation through homologous interactions or through Toll-like receptor (TLR) signals to regulate cytokine production and initiate a cellular response [16]. Some studies have reported that VLPs coated with DNA sequences rich in unmethylated CG motifs (cytosine phosphoguanosine oligodeoxynucleotides or “CpGs”) that are specific to APCs can not only reduce the serious side effects of naked CpGs but improve their pharmacokinetic efficiency. For example, when delivered in mice, VLPs enhance the adjuvant properties of CpGs and induce a stronger immune response [17]. Furthermore, some VLPs, like Qβ-VLPs, can induce T cell-independent immunoglobulin M (IgM) responses and T cell-dependent immunoglobulin G (IgG) responses [18].

### 2.2. Epitope-Displaying VLP Nanocarriers

Nanoparticles are widely used in materials science, medicine, and biology. VLPs are regarded as a type of nanoparticle because they structurally allow for the insertion of foreign genes or peptide fragments for antigen presentation. The repetitive sites on the surface of VLPs can be modified or coupled with foreign antigens of other pathogens by chemical or genetic fusion to form chimeric VLPs, which substantially enhance the stimulation of the immune system by the foreign antigens [19,20]. Foreign antigens displayed on the VLP surface can be achieved by gene engineering or chemical cross-linking. One way is to insert foreign antigens into some specific sites within the viral structural protein by fusion protein expression [21], the other way is to chemically decorate the VLP surface with target antigens through bifunctional cross-linkers, such as 4-(N-maleimidemethyl) loops Hexane-1-carboxylic acid sulfosuccinimide ester (sulfo-SMCC), nanogels, and so on [22]. Besides, sometimes it is necessary to encapsulate genes or small molecular substances inside the VLPs for effective delivery. Many VLPs can undergo transitions that lead to the formation of pores, thus the inner facet of VLPs can be exploited by a simple diffusion through their pores or a disassembly/reassembly process [23]. For example, doxorubicin (DOX) mixed with polyacrylic acid polymer can be packaged inside hepatitis B core antigen (HBcAg) VLPs by a dissociation and association process [22].

Numerous VLPs (e.g., adenovirus, hepatitis B virus, cowpea mosaic virus, rabbit hemorrhagic virus, among others) have been fused with viral structural proteins for the display of foreign antigens [24]. A frequent example is HBcAg, a hepatitis B virus core antigen that can self-assemble into icosahedral particles, and can be used as a therapeutic vaccine against chronic hepatitis B (CHB) or as a carrier for the presentation of various epitopes [25]. Indeed, recent work shows that HBc VLPs can display human cytomegalovirus AD-4 antigens with different densities on their surface epitopes in association with transpeptidase sortase A. When the densities reach above 64.2%, the VLPs can induce quite high antibody titers of AD-4 antigens [26]. Similarly, HBcAg VLPs have been used as an immunological vector for the influenza A virus (IAV) vaccine, Tandiflu1. Four conserved antigens of the extracellular domain of the IAV matrix protein 2 and hemagglutinin stems are inserted into the major immunodominant region (MIR) of HBcAg, which induces cross-reactivity and protective antibodies. Importantly, Tandiflu1 offers 100% protection against fatal doses of H1N1 IAV [27]. Finally, others have shown that VLPs from the RNA bacteriophage AP205 can simultaneously display the placental malaria antigen (VAR2CSA) and HPV L2 RG1 epitopes in high densities to establish a VLP-based combination vaccine against both placental malaria, caused by *Plasmodium falciparum* infection, and human papillomavirus [28]. In summary, VLPs have natural advantages as nanocarriers and are expected to be beneficial tools for the development of preventive or therapeutic vaccines.

### 2.3. VLP Delivery Platform

There are significant advantages of using VLPs as delivery platforms, including specific targeting, efficient host cell penetration, biocompatibility, and degradability [29]. Similar to natural viruses, receptors exposed on the surfaces of VLPs can mediate specific host cell targeting, and aid in facilitating endocytosis to efficiently penetrate the host cells [30]. Moreover, proteolytic mechanisms mean that VLPs can be degraded, and the degradative products are biocompatible. Therefore, VLPs can be used to envelope drugs, small molecules, and nucleic acids for targeted delivery, and could be harnessed for the development of gene therapy vectors. The bacteriophage MS2 VLPs can self-assemble into a VLP of T = 3. Some exogenous RNA operons can trigger the assembly of MS2 VLPs to package drugs, then be delivered to specific cells which is mediated by some specific guiding molecules that have been covalently conjugated to VLPs. [31,32]. Similarly, the receptor for vitamin folic acid (FA) is overexpressed on tumor cells. Covalently linking FA to the surface of cowpea mosaic virus(CPMV)and Hibiscus chlorotic ringspot virus (HCRSV) VLPs, can achieve cell-specific delivery [33]. In addition, conjugation C60 to bacteriophage Qβ-VLPs can significantly increase the solubility of C60 and promote the delivery to target cells [34].

Adenovirus (Ad) is the most commonly used viral vector for drug delivery or gene therapy [35]. The structural proteins of different Ad serotypes can self-assemble into VLPs of different sizes. The adenovirus dodecahedron (Dd) VLP is derived from serotype 3. Dd is used as a delivery vehicle because it can effectively cross cell membranes for the direct delivery of anticancer agents covalently attached to the surface of the carrier or as a promoter for drug delivery to tumors both in vivo and in vitro. For instance, Dd can successfully deliver oncogene inhibitors (e.g., a cap structure analog of eIF4E inhibitor and doxorubicin) to inhibit the proliferation of cancer cell cultures in vitro, reduce tumor growth by 40% in rat models of hepatocellular carcinoma (HCC), and significantly reduce the expression levels of oncogenes eIF4E and c-myc in rat tumor biopsies [36]. The use of Dd as a delivery agent can also help to reduce the cytotoxicity of anticancer agents by lowering the required effective dose. For instance, Dd-based delivery of bleomycin (BLM), an anticancer antibiotic, can induce cell death in transformed cells via double-strand DNA breakage and does so as efficiently as free bleomycin (BLM) but at a 100-times lower dose [37]. VLPs thus have the potential to significantly improve the bioavailability of a range of delivered substances.

## 3. VLP-Based Vaccines

At present, 110 viral proteins derived from 35 viral families have been shown to assemble into VLPs [38]. From this research, several VLP-based vaccines have been marketed for use in humans, including Recombivax HB and Engerix-B for hepatitis B virus (HBV), Gardasil, Cervarix, and Gardasil-9 for human papillomavirus (HPV), and Hecolin for hepatitis E virus (HEV). Several newer vaccines are also at various stages of the design, production, and approval process (Table 1).

### 3.1. Recombinant HBV Vaccine

Hepatitis B virus (HBV), an enveloped DNA virus grouped in the Hepadnaviridae family, is the main causative pathogen of hepatitis B [81]. The infection of HBV can lead to both acute and chronic hepatitis and substantially increase morbidity and mortality rate in the affected individuals [82]. According to epidemiological statistics, two billion people have hepatitis B serological characteristics worldwide, of whom 350 million have chronic hepatitis B [83]. Currently, vaccination is the most effective way to prevent HBV infection, with vaccines based on the self-assembly of the HBV surface antigen HBsAg into VLP particles.

To date, three generations of VLP vaccines against HBV have been developed. The first generation of vaccine was Heptavax-B, a hematogenous HBV vaccine composed of hepatitis B surface antigen particles (HBsAg VLP; diameter ~22 nm) isolated from the blood samples of patients with hepatitis B [84]. It was essentially an inactivated vaccine manufactured by human plasma that was collected from asymptomatic individuals who were suffering with chronic hepatitis B. Because HBsAg and HBV are both present in the plasma of the donor, it needs to be polished and inactivated by ultracentrifugation, pepsin digestion, urea, and formaldehyde treatment to obtain higher purity HBsAg [39,40]. Due to uncertainties around safety and source, its use was gradually replaced by second-generation vaccines, Recombivax HB and Engerix-B. These second-generation HBV vaccines, produced by Merck and GlaxoSmithKline respectively, are genetically engineered vaccines based on HBV VLPs [43]. Both use the *Saccharomyces cerevisiae* system to stably express HBsAg, producing particles about 20 nm in size and with a regular octahedral symmetrical structure. These vaccines are regarded as safer and more immunogenic than the first-generation, blood-borne hepatitis B vaccine, and continue to be widely used today [43]. The third-generation vaccine, Sci-B-Vac, contains three HBV surface antigens, including S, pre-S1, and pre-S2 antigens, and is expressed in mammalian Chinese hamster ovary (CHO) cells. Compared with the VLPs produced by the yeast system, which comprises only non-glycosylated HBsAg, these VLPs, with a mixture of glycosylated and non-glycosylated HBsAg, have higher immunogenicity [26]. Furthermore, the vaccine produces high titers of anti-HBsAg antibodies and protective antibodies against pre-S1 and pre-S2, with good efficacy and safety. It was approved for use in Israel and East Asia in 2017.

Recently, a new HBV vaccine, Heplisav-B, has been approved for use in adults in the United States. This vaccine uses the same 20 nm VLP self-assembly from HBsAg as the former three generations of vaccines, but also uses a CpG sequence 1018 as an adjuvant. Inoculation with two dosages within the month leads to better effect than the three-dosage Engerix-B within six months. Moreover, its immunogenicity is higher than that for Engerix-B. Therefore, Heplisav-B is expected to become a new generation of vaccine against HBV infection [49,82].

In addition to these preventive vaccines, VLP-based therapeutic hepatitis B vaccines are also being developed. In 2019, one group showed that multiple copies of the HBsAg-aa113-135 (SEQ13) polypeptide epitope could be displayed on a novel immuno-enhanced VLP carrier (CR-T3) derived from the roundleaf bat HBV core antigen (RBHBcAg). This core antigen could induce a specific antibody response mediated by HBV/HBsAg clearance to achieve a therapeutic effect, and is currently undergoing further research [85].

### 3.2. HPV VLP Vaccines

HPV is a circular, double-stranded DNA virus, and persistent HPV infection is a major cause of cervical cancer and genital warts [86]. At present, the four marketed prophylactic HPV vaccines are based on self-assembled VLPs that contain only the L1 protein [87,88]: Gardasil (Merck), Cervarix (GSK), Gardasil-9 (Merck) and Cecolin (Innovax). The L1 protein is the major structural protein of the virus, which can self-assemble into VLPs with high immunogenicity, and is capable of inducing a strong type-specific immune response [89]. Structural analysis shows that each HPV VLP consists of 72 L1 pentamers (five copies of the L1 protein), with a high induced immune response [90]. The quadrivalent vaccine Gardasil (Merck), which was licensed for use in the United States in 2006, mainly targets HPV 6/11/16/18 and provides 100% protection against virus infection caused by these four HPV types [53,54]. In 2009, a bivalent vaccine, Cervarix (GSK), was approved by the Food and Drug Administration (FDA). This vaccine used the baculovirus insect cell expression system, and mainly targets HPV16/18 [55,56]. Later, in 2014, Merck further launched the nine-valent vaccine, Gardasil-9, which provides added protection against HPV31/33/45/52/58 viruses in addition to HPV 6/11/16/18 provided by Gardasil. However, to induce antibody levels comparable with Gardasil, Gardasil-9 has a higher L1 antigen and aluminum adjuvant content than Gardasil [57]. Among the three vaccines, Cervarix has the lowest antigen concentration yet also offers better immunogenicity and long-term protection against HPV16 and 18. This is because Cervarix has a novel AS04 adjuvant system that carries the TLR4 agonist monophosphoryl lipid A (MPL), which directly stimulates antigen-presenting cells (APCs) [6,91].

In most developing countries, vaccine production via eukaryotic expression systems renders the vaccine unaffordable. Recently, a recombinant HPV type 16/18 bivalent vaccine (Cecolin) was developed using the *Escherichia coli* expression system, which could greatly reduce the cost of vaccine production. This system has excellent safety and efficacy in phase 3 clinical trials [58,59], producing the comparable levels of neutralizing antibodies as that of marketed vaccines and has recently been approved for marketing in China. Further, a nine-valent vaccine developed with *E. coli* has entered into phase 2 clinical trials [60].

More recently, it was proposed that VLPs could be used to express multiple antigens to offer protection against multiple strains of the same virus type through chimeric design. To this end, an “anti-multiple” single-particle chimeric vaccine was designed. The authors showed that the L1 loop region is closely related to HPV-type specificity, and that homologous substitution of the L1 loop region with that of a closely phylogenetically related loop region can offer cross-protection. A triple-type chimera of HPV33/58/52 VLPs can induce neutralizing titers that are comparable to a mixture of three wild-type VLPs, and this occurs both in mice and non-human primates. Based on this strategy, the authors are at present developing seven anti-20 type-cross vaccines, which are expected to prevent and control more comprehensively HPV-related diseases (e.g., cervical cancer and condyloma acuminata) [58,92].

### 3.3. Recombinant HEV Vaccine

The hepatitis E virus (HEV) VLP-based vaccine, Hecolin, launched in China in 2011, has brought significant benefits to the prevention of HEV infection [93]. HEV, a non-enveloped, single-stranded RNA virus, is the agent of enterically transmitted hepatitis with a worldwide distribution, and may cause severe acute hepatitis [94]. The genome length of HEV is 7.2 kb, and it includes three open reading frames [94], among which ORF2 encodes the only structural protein, pORF2. pORF2 is 660 amino acids in length and contains the core antigenic region (aa 458–607). The Hecolin vaccine was developed based on 20 to 30 nm diameter VLPs of a truncated version of pORF2 (aa 368–606), referred to as p239 [61]. Clinical trials have demonstrated that Hecolin has high immunogenicity and protection efficacy, and can induce high titers of HEV antibodies. It can effectively prevent hepatitis E infection within 4.5 years, with a protection rate of 93% [62,95]. Moreover, because the VLPs are prepared using the *E. coli* expression system, this largely reduces the cost associated with vaccine production and thus would be beneficial for vaccination in developing countries. At present, it is the only vaccine that can effectively prevent hepatitis E [61]. Besides Hecolin, two other HEV vaccines—rHEV vaccine (aa 112–607 aa) and p179 vaccine (aa 439–617)—have entered into clinical trials. The rHEV vaccine (GSK) consists of a 56-kD peptide expressed in insect cells, and phase 2 trials show a protection rate of 95.5% [63]. Meanwhile, the p179 vaccine, which is expressed in the *E. coli* system, is still in phase 1 clinical trials [64].

## 4. Research Progress on VLP-Based Vaccines in Development

VLP-based vaccine design has been extrapolated for the treatment of other viruses and pathogens. Many of these vaccines are still in experimental or development stages or still in clinical trials, some proving to be more complicated than others. For example, the development of a universal vaccine to achieve effective prevention against influenza has proven to be difficult. The influenza viruses have high genetic variation and exist as multiple types/subtypes. As such, in recent years, the design of multivalent chimeric VLP-based influenza vaccines has become a hot topic in influenza research. Recently, Novavax described the design of a nano-VLP influenza vaccine, tNIV, which is highly immunogenic and shown to be safe in clinical trials, and is capable of eliciting antibodies that target multiple broadly cross-reactive epitopes on hemagglutinin, covering a variety of conserved antigenic sites, to better circumvent the seasonal antigenic drift [65,66]. Another vaccine against the H5N1 influenza strain is also based on VLPs (Medicago) and has entered into phase 2 clinical trials [68].

Mosquirix (RTS,S/AS01) is considered to be the first advanced VLP vaccine targeting the pre-erythrocytic stage of the *Plasmodium falciparum* parasite, which causes malaria in humans via mosquitos. The antigen in the RTS,S/AS01 vaccine consists of a recombinant form of circumsporozoite protein (CSP) linked to HBsAg(S) via an amino acid repeat and T-cell epitope domain. These proteins can spontaneously enter the cells to form VLPs that subsequently serve as a vaccine [96]. In phase 3 clinical trials, this vaccine has been shown to reduce (~50%) the incidence of severe malaria in children between the ages of 5 and 17 months, with good safety and efficacy [69]. The vaccine can also induce moderate levels of CD4+ T cells and antibodies against CSP, a secreted protein of the sporozoite stage of the parasite [97]. Another VLP-based malaria vaccine, R21, which is formed from a single CSP-HBsAg fusion protein, has been shown to be immunogenic even in very low doses in mice, and has now entered phase 1/2a clinical trials [70].

In addition to VLPs containing antigens against a single viral strain, “consensus VLPs” that were designed with multiple antigens from different strains, can be used with the goal to potentially induce cross-protective immunity. The GII.4 consensus norovirus VLP, engineered from sequences of three genetically distinct naturally occurring GII.4 strains, can induce cross-reactive immune responses against different clusters of GII.4 noroviruses when immunization to animals by the intramuscular route with aluminum hydroxide as adjuvant [98]. Currently, a bivalent VLP-based vaccine consisting of the consensus GII.4 VLP and a GI.1 VLP is in phase 2 clinical trial. Formulated with the adjuvant containing MPL and aluminum hydroxide and administered intramuscularly, this vaccine showed good tolerance, safety, and reactogenicity in the clinical trials [99]. In addition, in order to cope with the diversity of the H5N1 influenza virus, a chimeric hemagglutinin (HA)-based vaccine was developed by a methodology termed computationally optimized broadly reactive antigen (COBRA), named Human COBRA 2. The Human COBRA 2 incorporated key neutralizing epitopes of two HA antigens, Human COBRA 2 (Hu-CO) and Human-Avian COBRA 2 (Hu-Av CO), was confirmed to be able to elicit broadly protective antibodies against heterologous clades of viruses [100].

Finally, prophylactic and therapeutic vaccines targeting different viruses, such as rotavirus, norovirus, and enterovirus, have evolved with the maturity of VLP vaccine technology (Table 1). These new vaccines are expected to prevent more human diseases [76,101,102].

## 5. Other VLP-Related Information

### 5.1. The Administration Route of VLP-Based Vaccine

The administration route is a fundamental parameter to be considered because it influences the stability of VLPs and the pharmacological outcome. Current vaccination methods include oral, intramuscular, intradermal, and intranasally, etc. [103,104,105]. According to Table 1, we find that almost all VLP-based vaccines are injected intramuscularly, and the injection site is mostly in the deltoid muscle of the upper arm. Studies have shown that compared with intramuscular injection of the gluteal muscle or subcutaneous injection of the arm, intramuscular injection of the Merck Sharp and Dohme (MSD) hepatitis B vaccine into the deltoid muscle of the upper arm induces the highest seroconversion rate and the highest average anti-HBs titer [106]. Adipocytes lack antigen-presenting cells, while there are fewer adipocytes in the deltoid muscle, which is conducive to antigen presentation to T or B cells [107]. In addition, studies have shown that the intradermal influenza vaccine can produce an immune response that is as effective as an intramuscular injection with less amount of immunogen [108]. Therefore, the appropriate route of administration should be determined according to the nature of different vaccine preparations and the purpose of vaccination.

### 5.2. VLP-Based Vaccine versus Virosomes-Based Vaccine

Virosomes, a spherical, unilamellar vesicles with a mean diameter of ~150 nm, and without nucleocapsid and genetic material of the source virus, are also used as important delivery tools in vaccine development [109,110]. Similar to VLPs, virosomes have repetitive epitopes and special envelope vesicle structure, can activate the immune response [111], and serve as vectors to display foreign antigens that are fused on the surface or inserted into a double bilayer [110]. Additionally, virosomes can be used as vaccine adjuvants to enhance the immunogenicity. A virosome-formulated vaccine, Epaxal, based on formalin-inactivated HAV (strain RG-SB), has proven that in the absence of aluminum hydroxide adjuvant, a single injection is well-tolerated and highly immunogenic, with 88%–97% of seroprotection [112]. Therefore, virosomes are also a promising vaccine adjuvant or delivery vehicle.

## 6. Discussion and Conclusion

VLPs are widely used in drug delivery, gene therapy, and vaccine development. As an immunogen, they are suitable in size and have a morphological structure similar to that of natural viruses, and can therefore induce a stronger immune response. As a nanocarrier, VLPs not only display foreign antigens, but can also be modified to express chimeras through chemical or genetic fusion technology, offering cross-neutralization protection against different viral strains. Finally, the targeted delivery of small molecule drugs and nucleic acids through VLPs improves the bioavailability of the delivered substance. However, it should not be overlooked that, although dozens of VLPs have been successfully prepared in the laboratory stage, there are still many problems to be solved in the development of effective vaccines and carriers. Indeed, it remains to be determined how to improve the expression of VLPs to meet the scale-up production process, how to effectively assemble VLPs, and how to reduce the immune response caused by VLPs when used as a delivery vehicle. Continued advances in VLP vaccine design will provide new possibilities for the treatment of numerous viral infections, and offer powerful tools for biomedical and vaccination research.

## Figures and Tables

**Figure 1 vaccines-08-00139-f001:**
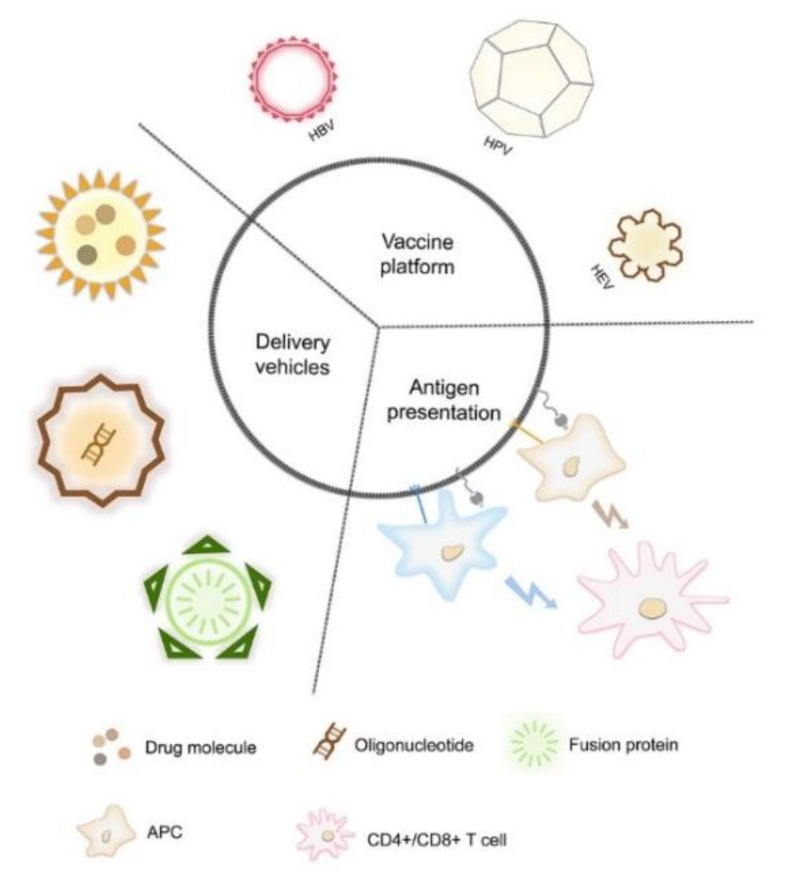
An overview of virus-like particles (VLPs) used as efficient nanocarriers for antigen presentation, cargo delivery, and as a vaccine platform.

**Figure 2 vaccines-08-00139-f002:**
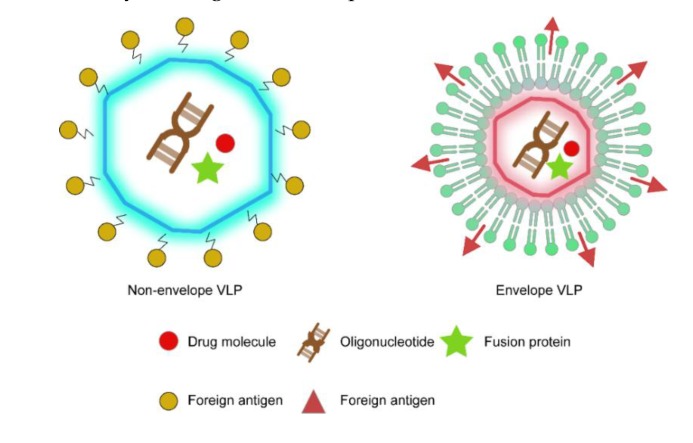
Non-enveloped and enveloped VLPs as a platform for antigen presentation and cargo delivery. The internal cavity of the VLPs accommodates exogenous oligonucleotides (rendered as brown gene-like patches), drug molecules (red circle), and small proteins (green pentagram) that can be delivered after the VLPs get the target organs or cells. The outer surface of the VLPs displays foreign proteins that are fused (red triangle) or covalently conjugated (yellow-green circle) to the particles.

**Table 1 vaccines-08-00139-t001:** VLP-based vaccine on the market or in clinical development.

Vaccine	Expression System	Adjuvant	VLP Platform	Antigen	Administration Route	Development Stage	References
Heptavax-B (Merck & Co.)	None (Plasma derived)	Aluminum hydroxide	HBsAg	SHBs	IM	Licensed	[39,40]
Engerix-B® (GSK, Belgium)	Yeast (*S. cerevisiae*)	Aluminum hydroxide	HBsAg	SHBs	IM	Licensed	[41,42,43]
Recombivax HB (H-B-Vax®II) (Merck & Co., USA)	Yeast (*S. cerevisiae*)	Aluminum sulfate	HBsAg	SHBs	IM	Licensed	[43,44]
Sci-B-Vac® (Bio-Hep-B®) (SciGen, Israel)	Mammalian (CHO cells)	Aluminum hydroxide	HBsAg	SHBs, MHBs, LHBs	IM	Licensed	[45,46,47,48]
Heplisav-B (Dynavax)	Yeast (*H. polymorpha*)	1018 ISS	HBsAg	SHBs	IM	Licensed	[49,50]
Fendrix® (GSK, Belgium)	Yeast (*S. cerevisiae*)	AS04 (Aluminum hydroxide & MPL)	HBsAg	SHBs	IM	Licensed	[51]
Hepavax-Gene® (Crucell)	Yeast (*H. polymorpha*)	Aluminum hydroxide	HBsAg	SHBs, MHBs	IM	Licensed	[41,52]
Gardasil® (Merck & Co.)	Yeast (*S. cerevisiae*)	Aluminum hydroxyphosphate sulphate	HPV	HPV6/11/16/18 VLP	IM	Licensed	[53,54]
Cervarix® (GSK)	Insect cells (High FiveTM cells)	AS04 (Aluminum hydroxide & MPL)	HPV	HPV16/18 VLP	IM	Licensed	[55,56]
Gardasil-9® (Merck & Co.)	Yeast (*S. cerevisiae*)	Aluminum hydroxyphosphate sulfate	HPV	HPV6/11/16/18/31/33/45/52/58 VLP	IM	Licensed	[57]
Cecolin® (Innovax)	Bacteria (*E. coli*)	Aluminum hydroxide	HPV	HPV16/18 VLP	IM	Licensed	[58,59]
Cecolin-9® (Innovax)	Bacteria (*E. coli*)	Aluminum hydroxide	HPV	HPV6/11/16/18/31/33/45/52/58 VLP	IM	Phase 2	[60]
Hecolin® (Innovax)	Bacteria (*E. coli*)	Aluminum hydroxide	HEV	HEV p239 (aa 439–617)	IM	Licensed	[61,62]
rHEV (GSK)	Insect cells	Aluminum hydroxide	HEV	HEV (aa 112–607)	IM	Phase 2	[63]
p179 (Changchun Institute of Biological Products Co. Ltd.)	Bacteria (*E. coli*)	Aluminum hydroxide	HEV	HEV (aa 439–617)	IM	Phase 1	[64]
tNIV (Novavax)	Insect cells (Sf-9 cells)	Matrix-M	Influenza virus	A/Hong Kong/4801/2014(H3N2) HA	IM	Phase 1/2a	[65,66]
gH1-Qbeta	Bacteria (*E. coli*)	None	Qβ-VLPs	A/California/07/2009(H1N1) HA(gH1 domain)	IM	Phase 1	[67]
H5VLP+GLA vaccine	Plant	GLA-SE	Medicago VLP	A/Indonesia/05/2005(H5N1) H5	IM	Phase 2	[68]
RTS,S/AS01 (GSK)	Yeast (*S. cerevisiae*)	AS01	HBsAg	P.*f*.CSP	IM	Phase 3	[69]
R21	Yeast (Pichia pastoris yeast strain)	Abisco-100 and Matrix-M	HBsAg	P.*f*.CSP	IM	Phase 1/2a	[70]
Norovirus VLPs and rotavirus VP6 combined vaccine	Insect cells	None	Norovirus GII-4 VLP	NoV GII-4 VLP, rVP6	IM/ID	Pre-clinical	[71,72]
P2-VP8-P [8] vaccine (Walter Reed Army Institute of Research)	Bacteria (*E. coli*)	Aluminum hydroxide	Wa strain (G1P [8]) of human rotavirus	VP8 subunit (aa 64–223) & P2 epitope from tetanus toxin	IM	Phase 1	[73]
Rotaviruses VP 2/6/7 and VP 2/6 (Baylor College of Medicine, USA)	Insect cells	None or cholera toxin	Rotavirus VLPs	VP 2/6/7 and VP 2/6	Nasally immunized	Pre-clinical	[74]
Bivalent GI.1/GII.4 vaccine (Takeda Pharmaceutical Company Limited)	Insect cells (Sf-9 cells)	Aluminum hydroxide & MPL	Norovirus VLPs	GI.1VLP, GII.4 VLP	IM	Phase 2b	[75]
EV71 VLP C4a	Insect cells (High FiveTM cells)	Aluminum hydroxide	EV 71 VLP	EV71 C4a-gp41	IM	Pre-clinical	[76]
HFMD tetravalent vaccine	Insect cells	Alhydrogel adjuvant	EV71, CVA6, CVA10, CVA16 VLP	EV71, CVA6, CVA10, CVA16 VLP	Intraperitoneally	Pre-clinical	[77]
RSV F (Novavax)	Insect (Sf-9 cells)	Aluminum phosphate	RSV	RSV-F	IM	Phase 1	[78,79]
Rabies (TJU)	Plant (Transgenic spinach)	None	AlMV	Rabies GP/NP	Oral	Phase 1	[80]

* Abbreviations: HBsAg: hepatitis B surface antigen; SHBs: small S hepatitis B surface antigen; MHBs: middle Pre-S2 HBV peptide; LHBs: large Pre-S1 envelope protein; IM: intramuscular injection; GSK: GlaxoSmithKline; *S. cerevisiae*: *Saccharomyces cerevisiae*; ID: intradermally; CHO cells: Chinese hamster ovary cells; *H. polymorpha*: *Hansenula polymorpha*; ISS: immunostimulatory sequences; HPV: human papillomavirus; *E.*
*coli*: *Escherichia coli*; HEV: hepatitis E Virus; Sf-9 cells: *spodoptera frugiperda* cells; H3N2,H1N1,and H5N1: a subtypes of influenza viruses; HA: hemagglutinin; Qβ-VLPs: RNA bacteriophage Qbeta VLPs; GLA-SE: glucopyranosyl lipid adjuvant-stable emulsion; H5: the A/Indonesia/05/2005 H5N1 influenza hemagglutinin protein; P.*f*.: *Plasmodium falciparum*; CSP: circumsporozoite protein; VP2/6/7/8: rotavirus capsid protein; MPL: monophosphoryl lipid A; EV: enterovirus; CV: coxsackieviruses; HFMD: hand, foot, and mouth disease; RSV: respiratory syncytial virus; RSV-F: the F protein ectodomain of respiratory syncytial virus; AIMV: alfalfa mosaic virus; GP: glycoprotein; NP: nucleoprotein.

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
