# Peer review of "Recent Progress on the Versatility of Virus-Like Particles"

_vaccines, 2020, doi:10.3390/vaccines8010139_

Round 1

Reviewer 1 Report

This manuscript is a concise review on the applications of virus-like particles (VLPs) as modulators of the immune system and as drug delivery platforms. The manuscript also summarizes the current approved VLP-based vaccines, while glancing over VLP-based vaccines still in development. I believe that this manuscript should be published with major revisions, as the topic of VLP-based technologies is becoming more popular. However, while the information presented on current vaccines is detailed, more emphasis should be put on VLP-based vaccines in development/clinical trials. Importantly, many of the citations in this manuscript were incorrect/irrelevant and should be revised. As mentioned in major comment 1, more primary articles should be cited, as summarizing data from various primary sources and drawing conclusions from them is one of the main purposes of writing a review article. Thus, I do not believe this manuscript should be published until those changes, and the ones mentioned in the major comments below, have been made.

Major Comments:

1) The authors should cite more primary research articles over review articles. Currently, it seems to be that ~40% of the articles cited are review articles. Citing the occasional review article is okay, especially in the introduction or if the review article goes into more depth on a particular subject, however primary articles containing the actual data are often more relevant.

2) Please change Section 3 to "VLP-based vaccines" and add a Section 4 "Research progress on VLP-based vaccines in development" (or something like that), as it warrants its own section. Consequently, more details should be added to VLP-based vaccines that are currently in development, such as the multitude of influenza VLP-based vaccine candidates. How about VLP-based vaccines for norovirus, rotavirus, and enterovirus? Those are glossed over in two sentences (Lines 230-232), but the authors should add more information on those three viruses/VLP-based vaccines currently being developed. The norovirus VLP-based vaccine, for example, has data published from Phase II clinical trials that could be summarized here. Please also discuss plant-based VLPs (TMV chimeric particles), RSV, and Dengue. There are also chimeric HBV/HCV VLPs in the works. All of this additional information will help this manuscript stand out, as there are a multitude of review articles that discuss VLPs and their applications, but not many recent articles that review the current VLP-based vaccine candidates.

3) The authors should mention and expand upon the concept of "Consensus VLPs" as a vaccine candidate in new Section 4 (see major comment 2). See papers by Ted Ross, for example [Nunez et al. 2020, Vaccine], and also [Parra et al. 2012, Vaccine] for influenza and norovirus, respectively.

4) Please add more details about the VLP-based vaccines against malaria. What VLPs are used and what malaria antigens are expressed on the VLPs, etc?

5) Please discuss in more detail the ability of VLPs to target certain tissues for efficient drug delivery platforms, as that is one of the major benefits of using VLPs, and one of the reasons why required doses are lower compared to administering the drug alone. The subject is briefly mentioned, but applied examples and a more in-depth discussion would be beneficial to the manuscript.

6) Line 27: The authors cite a review article that describes adjuvants in VLP-based vaccine formulations. The citation should be changed to more accurately represent the sentence "vaccination is one of the most cost-effective ways to control and prevent the spread of infectious disease". Perhaps an epidemiological study or a cost-benefit modeling study on vaccination rates and reduction of disease/disease burden would be a better choice.

7) Lines 31-34: Please tone down this sentence. Many approved, non-VLP-based vaccines are well-tolerated and effective in humans, such as MMR, inactivated poliovirus (IPV), and the seasonal influenza vaccines. The sentence can be modified to (for example): "Although many traditional vaccines are safe and effective in humans, several inherent risks are associated with certain live attenuated and inactivated vaccines, including..."

In addition, citation [3] and citation [4] do not mention ineffectual protection, incomplete attenuation, unstable toxicity, and risks of mutating back to the virulent form for "traditional vaccines". Thus, they are not good citations for Lines 31-34. I would recommend citing papers that describe TC83 (a live-attenuated vaccine against VEEV) and/or the oral poliovirus vaccine (OPV), both which readily can mutate back to the virulent form.

8) Lines 120-121: Citation [26] appears to be incorrect. Please update.

9) I cannot find citation [2], "The birth and progression of vaccines" in https://www.ncbi.nlm.nih.gov/pubmed or by a simple Google search. I recommend citing an easier-to-access article.

10) Line 40: Citation [5] is incorrect, as it is a review on antigen presentation in B cells and does not contain information on VLPs and their lack of genetic material, pathogenicity, or infectivity. Please remove or change.

11) Line 41: Please change citation [6] to something more relevant. For example, see [Jiang et al. 1992, Journal of Virology], [Deschuyteneer et al. 2010, Human Vaccines], [Metz et al. 2018, Virology Journal], [Wu et al. 2010, Plos One], [Roldao et al. 2010, Expert Review of Vaccines], etc. Some of these citations could also be used for Line 59 instead of citation [8].

12) Line 59: Please find better citation instead of the current citation [9]. For example, see [Mohsen et al. 2018, Vaccines].

13) Please change citation [23] to something more relevant.

14) Line 117: Please cite the primary article, not the review.

15) Lines 138-139: Please cite the sentence beginning with "These vaccines are regarded as safer and more immunogenic..."

16) Line 165: Please cite primary research articles instead of [35].

17) Line 198: Please change citation [46] to something more relevant, or remove it.

18) Line 201: Citations [47] and [48] can be replaced with [49], or by citing [Li et al. 2005, Vaccine].

19) Please cite Lines 208-209.

Minor comments:

1) Line 28: Please change to "...by artificial attenuation..."

2) Line 33: Please change to "....the inherent limitations...make it difficult..."

3) The wording of Lines 72-73 is confusing.

4) Line 79: Please change to "...for the display of foreign antigens"

5) Line 79: Please clarify what "cross-activating the immune system" means.

6) Line 94: Please italicize "Plasmodium falciparum", as it is a genus/species classification.

7) Citation [24] seems out of place.

8) Line 133-134: Please expand upon "It was essentially an attenuated vaccine..."

9) Similar to what you have written for HPV, a brief background about Hepatitis B and Hepatitis E disease could be added to sections 3.1 and 3.3 to emphasize the importance of developing vaccines for prevention of those diseases.

10) Line 216: Please change to "As such, in recent years..."

11) Line 220: Do you mean "...could effectively increase titers against influenza genetic drift variants"?

Author Response

Comments to the Author
Reviewer: This manuscript is a concise review on the applications of virus-like particles (VLPs) as modulators of the immune system and as drug delivery platforms. The manuscript also summarizes the current approved VLP-based vaccines, while glancing over VLP-based vaccines still in development. I believe that this manuscript should be published with major revisions, as the topic of VLP-based technologies is becoming more popular. However, while the information presented on current vaccines is detailed, more emphasis should be put on VLP-based vaccines in development/clinical trials. Importantly, many of the citations in this manuscript were incorrect/irrelevant and should be revised. As mentioned in major comment 1, more primary articles should be cited, as summarizing data from various primary sources and drawing conclusions from them is one of the main purposes of writing a review article. Thus, I do not believe this manuscript should be published until those changes, and the ones mentioned in the major comments below, have been made.
Response: We thank and agree with the reviewer for the summary on our study. We have modified the content of the article according to your suggestions, added more information about VLP, such as the composition of VLP, the technology of fusion of VLP and foreign antigens, details of vaccines that are already on the market or under development, and added some primary articles as reference to replace some of review articles cited in the previous version.

Specific comments
Comment 1:
The authors should cite more primary research articles over review articles. Currently, it seems to be that ~40% of the articles cited are review articles. Citing the occasional review article is okay, especially in the introduction or if the review article goes into more depth on a particular subject, however primary articles containing the actual data are often more relevant.
Response: As suggested, we have replaced most of the citations with more relevant primary articles, please find the replacement details in the compared version.

Comment 2: Please change Section 3 to "VLP-based vaccines" and add a Section 4 "Research progress on VLP-based vaccines in development" (or something like that), as it warrants its own section. Consequently, more details should be added to VLP-based vaccines that are currently in development, such as the multitude of influenza VLP-based vaccine candidates. How about VLP-based vaccines for norovirus, rotavirus, and enterovirus? Those are glossed over in two sentences (Lines 230-232), but the authors should add more information on those three viruses/VLP-based vaccines currently being developed. The norovirus VLP-based vaccine, for example, has data published from Phase II clinical trials that could be summarized here. Please also discuss plant-based VLPs (TMV chimeric particles), RSV, and Dengue. There are also chimeric HBV/HCV VLPs in the works. All of this additional information will help this manuscript stand out, as there are a multitude of review articles that discuss VLPs and their applications, but not many recent articles that review the current VLP-based vaccine candidates.

Response: As suggested, we have detailed the research progress of influenza and malaria VLP-based vaccines in Section 4 and give examples of the concept of consensus VLP. In addition, we provided a new table (Table 1) to show the formulations of VLP vaccine both on the market and under clinical development, including vaccine composition, administration route and their corresponding references.

Comment 3: The authors should mention and expand upon the concept of "Consensus VLPs" as a vaccine candidate in new Section 4 (see major comment 2). See papers by Ted Ross, for example [Nunez et al. 2020, Vaccine], and also [Parra et al. 2012, Vaccine] for influenza and norovirus, respectively.

Response: Thanks for providing information about “consensus VLP”, we added the information in section 4.

“In addition to the common VLP vaccine form, a new VLP candidate vaccine, named "Consensus VLP", has been used in the development of VLP-based vaccines for noroviruses and influenza viruses. The GII.4 consensus norovirus VLP, engineered from sequences of three genetically distinct naturally occurring GII.4 strains, can induced cross-reactive immune responses against different clusters of GII.4 noroviruses when immunization to animals by the intramuscular route with aluminum hydroxide as adjuvant [Parra, G. I., et al. 2012, Vaccine].

In addition, in order to cope with the diversity of the H5N1 influenza virus, a chimeric hemagglutinin (HA) antigen Human COBRA 2 that incorporated key neutralizing epitopes of two HA antigens, Human COBRA 2 (Hu-CO) and Human-Avian COBRA 2 (Hu-Av CO), was confirmed to be able to elicit broadly protective antibodies against heterologous clades of  viruses [Nunez et al. 2020, Vaccine].”

Comment 4: Please add more details about the VLP-based vaccines against malaria. What VLPs are used and what malaria antigens are expressed on the VLPs, etc?

Response: As suggested, we detailed the composition of the VLP-based vaccines against malaria. “The antigen in RTS,S/AS01 vaccine is a recombinant protein, which contains recombinant form of circumsporozoite protein (CSP),asparagine-alanine-asparagine-proline (NANP) amino acid repeat regions (R) and T-cell epitope domain (T) linked to HBsAg(S), These proteins can spontaneously enter the cells to form VLPs, subsequently serves as a vaccine [Moris, P., et al., 2018, Hum Vaccin Immunother].” And “Another VLP-based malaria vaccine, R21, which are formed from a single CSP-hepatitis B surface antigen (HBsAg) fusion protein…”

Comment 5: Please discuss in more detail the ability of VLPs to target certain tissues for efficient drug delivery platforms, as that is one of the major benefits of using VLPs, and one of the reasons why required doses are lower compared to administering the drug alone. The subject is briefly mentioned, but applied examples and a more in-depth discussion would be beneficial to the manuscript.

Response: As suggested, we detailed the ability of VLPs to target certain tissues for efficient drug delivery platforms in section 2.3 as following: “The bacteriophage MS2 VLPs can self-assemble into a VLP of T = 3. Covalently combining the drug to be delivered with the RNA operon, can induce the reassembly of MS2 VLPs, then package the drugs. These packaged drugs can then be directed to specific cells by further covalent decoration of the capsids with targeting molecules [Brown, W. L., et al. 2002, Intervirology],[Kovacs, E. W., et al. 2007, Bioconjug Chem]. Similarly, the receptor for vitamin folic acid (FA) is overexpressed on tumor cells, covalently linking FA to the surface of cowpea mosaic virus (CPMV) and Hibiscus chlorotic ringspot virus (HCRSV) VLPs, can achieved cell-specific delivery [Destito, G., et al. 2007, Chem Biol]. In addition, conjugation C(60) to bacteriophage Qβ-VLPs can significantly increase the solubility of C60 and promote the delivery to target cells [Steinmetz, N. F., et al. 2009, J Am Chem Soc].”

Comment 6: Line 27: The authors cite a review article that describes adjuvants in VLP-based vaccine formulations. The citation should be changed to more accurately represent the sentence "vaccination is one of the most cost-effective ways to control and prevent the spread of infectious disease". Perhaps an epidemiological study or a cost-benefit modeling study on vaccination rates and reduction of disease/disease burden would be a better choice.

Response: As suggested, we have cited a new review paper [Standaert B, et al. 2017, J Mark Access Health Policy] about the economic assessment of vaccines.

Comment 7: Lines 31-34: Please tone down this sentence. Many approved, non-VLP-based vaccines are well-tolerated and effective in humans, such as MMR, inactivated poliovirus (IPV), and the seasonal influenza vaccines. The sentence can be modified to (for example): "Although many traditional vaccines are safe and effective in humans, several inherent risks are associated with certain live attenuated and inactivated vaccines, including..."

In addition, citation [3] and citation [4] do not mention ineffectual protection, incomplete attenuation, unstable toxicity, and risks of mutating back to the virulent form for "traditional vaccines". Thus, they are not good citations for Lines 31-34. I would recommend citing papers that describe TC83 (a live-attenuated vaccine against VEEV) and/or the oral poliovirus vaccine (OPV), both which readily can mutate back to the virulent form.

Response: As suggested, lines31-34 were revised as: “Although many traditional vaccines are safe and effective in humans, several inherent risks are associated with certain live attenuated and inactivated vaccines, including incomplete attenuation, unstable toxicity, and risk of mutation...”

In addition, citation [3] and citation [4] have been replaced by [Tretyakova I., et al. 2020,vaccine] and [Tebbens, R. J., et al. 2006, Risk Anal].

Comment 8: Lines 120-121: Citation [26] appears to be incorrect. Please update.

Response: As suggested, we have cited a review paper [Jadwiga Chroboczek., et al. 2014, Acta Biochim Pol] about Virus-like particles as vaccine. It is reported at present, total 110 viral proteins derived from 35 viral families have been shown to be capable of assembling into VLPs.

Comment 9: I cannot find citation [2], "The birth and progression of vaccines" in https://www.ncbi.nlm.nih.gov/pubmed or by a simple Google search. I recommend citing an easier-to-access article.

Response: We apologize for this confusion. We have cited a new review paper [Vetter, V., et al. 2018, Ann Med].

Comment 10: Line 40: Citation [5] is incorrect, as it is a review on antigen presentation in B cells and does not contain information on VLPs and their lack of genetic material, pathogenicity, or infectivity. Please remove or change.

Response: We apologize for this. We have cited new review paper[Metz, S. W., et al. 2018, Virol J].

Comment 11: Line 41: Please change citation [6] to something more relevant. For example, see [Jiang et al. 1992, Journal of Virology], [Deschuyteneer et al. 2010, Human Vaccines], [Metz et al. 2018, Virology Journal], [Wu et al. 2010, Plos One], [Roldao et al. 2010, Expert Review of Vaccines], etc. Some of these citations could also be used for Line 59 instead of citation [8].

Response: As suggested, we have change citation [6] to [Deschuyteneer, M., et al. 2010, Human Vaccines], and change citation [8] to [Metz, S. W., et al. 2018, Virol J].

Comment 12: Line 59: Please find better citation instead of the current citation [9]. For example, see [Mohsen et al. 2018, Vaccines].

Response: As suggested, we have change citation [9] to [Mohsen et al. 2018, Vaccines].

Comment 13: Please change citation [23] to something more relevant.

Response: As suggested, we have change citation [23] to [Wold, W. S. and K et al. Toth 2013, Curr Gene Ther]

Comment 14: Line 117: Please cite the primary article, not the review.

Response: As suggested, we have change citation [24] to [Zochowska, M., et al. 2009, PLoS One].

Comment 15: Lines 138-139: Please cite the sentence beginning with "These vaccines are regarded as safer and more immunogenic..."

Response: As suggested, we have cited a review paper [Lacson, E., et al. 2005, Hemodial Int] for the sentence beginning with "These vaccines are regarded as safer and more immunogenic..."

Comment 16: Line 165: Please cite primary research articles instead of [35].

Response: As suggested, we have change citation [35] to [Chabeda, A., et al. 2019, Front Plant Sci]. In addition, amend this sentence to: “The L1 protein is the major structural protein of the virus, which can self-assemble into VLP with high immunogenicity, and is capable of inducing a strong type-specific immune response. Structural analysis…”

Comment 17: Line 198: Please change citation [46] to something more relevant, or remove it.

Response: As suggested, we have change citation [46] to [ Cai W , et al. 2016, Sci Rep].

Comment 18: Line 201: Citations [47] and [48] can be replaced with [49], or by citing [Li et al. 2005, Vaccine].

Response: As suggested, we have change citation [47] and [48] to [Li et al. 2005, Vaccine].

Comment 19: Please cite Lines 208-209.

Response: As suggested, we have cited a review paper [Shrestha, M. P., et al. 2007, N Engl J Med].

Minor comments:

Comment 1: Line 28: Please change to "...by artificial attenuation..."

Response: As suggested, we have changed “…by artificial attenuating…” to "...by artificial attenuation...".

Comment 2: Line 33: Please change to "....the inherent limitations...make it difficult..."

Response: As suggested, we have changed the sentence to "....the inherent limitations...make it difficult...".

Comment 3: The wording of Lines 72-73 is confusing.

Response: We apologize for this confusion. This sentence is intended to express that some VLPs can induce T cell-independent IgM responses and T cell-dependent IgG responses. We change the sentence to “Furthermore, some VLPs, like Qβ-VLPs, can induce T cell-independent IgM responses and T cell-dependent IgG responses.”

Comment 4: Line 79: Please change to "...for the display of foreign antigens"

Response: As suggested, we have changed the sentence to "...for the display of foreign antigens".

Comment 5: Line 79: Please clarify what "cross-activating the immune system" means.

Response: We apologize for this confusion. What we want to express is that VLP as a vector can stimulate the immune system and activate T or B cell responses, as well as foreign antigens can be displayed on VLPs. Therefore, VLP vector can enhance the stimulation of the immune system by foreign antigens[ Billaud, J. N., et al. 2005, J Virol] and [Peacey, M., et al. 2007, Biotechnol Bioeng]. We rephase the whole sentence as: "The repetitive sites on the surface of VLPs can be modified or coupled with foreign antigens of other pathogens by chemical or genetic fusion to form chimeric VLPs, which will substantially enhance the stimulation to immune system by the foreign antigens."

Comment 6: Line 94: Please italicize "Plasmodium falciparum", as it is a genus/species classification.

Response: As suggested, we have italicized "Plasmodium falciparum".

Comment 7: Citation [24] seems out of place.

Response: As suggested, we have changed citation [24] in line117 to [Zochowska, M., et al. 2009, PLoS One].And remove Citation [24] in line111.

Comment 8: Line 133-134: Please expand upon "It was essentially an attenuated vaccine..."

Response: As suggested, we expand the sentence to “It was essentially an inactivated vaccine manufactured by human plasma that was collected from asymptomatic individuals who were suffering with chronic hepatitis B. Because HBsAg and HBV are both present in the plasma of the donor, it needs to be polished and inactivated by ultracentrifugation, pepsin digestion, urea and formaldehyde treatment to obtain higher purity HBsAg. [Gerety, R. J. and E. Tabor ,1983, Jama] [ Szmuness, W., et al. 1980,N Engl J Med]. Due to…”

Comment 9: Similar to what you have written for HPV, a brief background about Hepatitis B and Hepatitis E disease could be added to sections 3.1 and 3.3 to emphasize the importance of developing vaccines for prevention of those diseases.

Response: As suggested, we have added relevant background about HBV and HEV.  “Hepatitis B virus (HBV), an enveloped DNA virus grouped in the Hepadnaviridae family, is the main causative pathogen of hepatitis B [Karayiannis P,2017, Hepatol Int]. The infection of HBV can lead to both acute and chronic hepatitis and substantially increase morbidity and mortality rate in the affected individuals[Splawn, L. M., et al. 2018, Drugs Today (Barc)]. According to…”

 “HEV, a non-enveloped, single-stranded RNA virus, is the agent of enterically transmitted hepatitis, with a worldwide distribution and may cause severe acute hepatitis[ Cai W , et al. 2016, Sci Rep].The genome length of HEV is 7.2 kb, it includes…”

Comment 10: Line 216: Please change to "As such, in recent years..."

Response: As suggested, we have changed the sentence to "As such, in recent years...".

Comment 11: Line 220: Do you mean "...could effectively increase titers against influenza genetic drift variants"?

Response: Sorry for the obscure information, the sentence has been rephrased as, “… tNIV, which is highly immunogenic and shown to be safe in clinical trials, and is capable of eliciting antibodies response that targets multiple broadly cross-reactive epitopes on hemagglutinin, covering a variety of conserved antigenic sites, to better circumvent the seasonal antigenic drift [ Portnoff, A. D., et al. 2020, Vaccines (Basel)].”

Reviewer 2 Report

The manuscript describes the various applications of virus-like particles (VLPs). The rationale was provided but specific aspects should be discussed in order to provide a complete panel of information to the readers.

The composition of VLPs should be better described. Are lipids/phospholipids used to obtain VLPs? A schematic representation can be useful.

Some info on the preparation procedure should be added. Moreover, the approaches used to modify the surface of VLPs should be discussed especially when a molecule/derivative is exposed. On the other hand, the techniques used to entrap molecules/active compounds in VLPs are required.  

A comparison with virosomes should be added (for example Epaxal - Bovier. Expert Rev Vaccines. 2008;7:1141-50).

The administration route is a fundamental parameter to be considered because it influences the stability of VLPs and the pharmacological outcome. The physico-chemical and technological properties of the various formulations are modulated as a function of the administration route. The authors should briefly discuss this point.

A table containing the VLP formulations on the market, their composition, the administration route and specific references should be added in order to improve the quality of the manuscript.  

Author Response

Reviewer: The manuscript describes the various applications of virus-like particles (VLPs). The rationale was provided but specific aspects should be discussed in order to provide a complete panel of information to the readers.

Response: We thank and agree with the reviewer for the summary on our study. We have modified the content of the article according to your suggestions, added more information about VLP, such as the composition of VLP, the technology of fusion of VLP and foreign antigens, details of vaccines that are already on the market or under development.

Specific comments
Comment 1:
The composition of VLPs should be better described. Are lipids/phospholipids used to obtain VLPs? A schematic representation can be useful.
Response: As suggested, we detailed the composition of the VLP in section 2 as following, and drew a schematic map (the new Figure 2).

“Free of genetic material inside, VLPs have broad potential applications and are a hot topic in many current research areas. In terms of structure architecture, VLP can be divided into two types: non-enveloped and enveloped. Non-enveloped VLPs, such as HPV VLPs, are relatively simple in structure and consist of a single or multiple structural proteins of a specific virus[Phelps, D. K., et al. 2000, Curr Opin Struct Biol] [Sasagawa, T., et al. 1995, Virology]. Enveloped-VLPs have a complex structure constituted by multiple components of the virus and host membrane components, the most prominent example of enveloped VLPs is influenza virus[Chen, B. J., et al. 2007, J Virol] [Haynes, J. R. 2009, Expert Rev Vaccines]. In addition of capability to stimulate an immune response, both forms of VLPs can be manipulated into a chimeric structure to display foreign antigens for a broader immunity, and can be repurposed as highly specific nanocarriers for the delivery of antigens or therapeutic molecules.”

Figure 2. Non-enveloped and enveloped VLPs as a platform for antigen presentation and cargo delivery.

Comment 2: Some info on the preparation procedure should be added. Moreover, the approaches used to modify the surface of VLPs should be discussed especially when a molecule/derivative is exposed. On the other hand, the techniques used to entrap molecules/active compounds in VLPs are required.

Response: Thanks for your instructive comments. As suggested, we outline how to display exogenous antigens on the surface of VLPs or encapsulate exogenous substances inside VLPs. The following was added in section 2.2.

“…One way is to generate recombinant fusion constructs, using genetic engineering techniques to insert foreign antigens into specific sites within the viral structural protein [Yong et al., 2015a,Appl Environ Microbiol], or through bifunctional cross-linkers, such as 4- (N-maleimidemethyl) loops Hexane-1-carboxylic acid sulfosuccinimide ester (sulfoSMCC) and nanogels chemically conjugate VLPs with the target antigen [Biabanikhankahdani et al., 2016,Sci Rep], so that the foreign antigen can be displayed on the surface of the VLPs. In addition to displaying exogenous antigens on the surface of VLPs, sometimes it is necessary to encapsulate genes or small molecular substances inside the VLPs for effective delivery. Many VLPs can undergo transitions that lead to the formation of pores, thus the inner facet of VLPs can be exploited by a simple diffusion through their pores or a disassembly/reassembly process[Goldinger, S. M., et al. 2012, Eur J Immunol]. For example, doxorubicin (DOX) was mixed with polyacrylic acid polymer can be packaged inside HBcAg VLPs by a dissociation and association process [ Biabanikhankahdani, R., et al. 2016, Sci Rep]. Numerous VLPs…”

Comment 3: A comparison with virosomes should be added (for example Epaxal - Bovier. Expert Rev Vaccines. 2008;7:1141-50).

Response: Thanks for providing information about “VLP-based vaccine vs virosomes-based vaccine”, we added the following in section 5.

“Virosomes, a spherical, unilamellar vesicles with a mean diameter of ~ 150 nm, and without nucleocapsid and genetic material of the source virus, are also used as important delivery tools in vaccine development [Clarke, P. D., et al. 2006, Travel Med Infect Dis], [Zurbriggen, R., et al. 2000, Prog Lipid Res]. Similar to VLPs, virosomes have repetitive epitopes and special envelope vesicle structure, can activate the immune response [Fehr, T., et al. 1997, J Exp Med], and serve as vectors to display foreign antigens that are fused on the surface or inserted into a double bilayer [Zurbriggen, R., et al. 2000, Prog Lipid Res]. Additionally, virosomes can be used as vaccine adjuvants to enhance the immunogenicity. A virosome-formulated vaccine, Epaxal®, based on formalin inactivated HAV (strain RG-SB), has been proven that in the absence of aluminum hydroxide adjuvant, a single injection is well-tolerated and highly immunogenic, with 88–97% of seroprotection [Bovier, P. A. 2008, Expert Rev Vaccines]. Therefore, virosomes are also a promising vaccine adjuvant or delivery vehicle.”

Comment 4: The administration route is a fundamental parameter to be considered because it influences the stability of VLPs and the pharmacological outcome. The physico-chemical and technological properties of the various formulations are modulated as a function of the administration route. The authors should briefly discuss this point.

Response: As suggested, we added the following sentences in section 5.

“The administration route is a fundamental parameter to be considered because it influences the stability of VLPs and the pharmacological outcome. Current vaccination methods include oral, intramuscular, intradermal and intranasally, etc. [Atmar R L, et al.2011, N Engl J Med], [ Bernstein D I,et al.2015, J Infect Dis], [Kim L, et al. 2018,JCI Insight]. According to Table 1, we find that almost all VLP-based vaccines are injected intramuscularly, and the injection site is mostly in the deltoid muscle of the upper arm. Studies have shown that compared with intramuscular injection of the gluteal muscle or subcutaneous injection of the arm, intramuscular injection of the Merck Sharp & Dohme(MSD) hepatitis B vaccine into the deltoid muscle of the upper arm induces the highest seroconversion rate and the highest average anti-HBs titer [de Lalla, F., et al. 1988, Eur J Epidemiol]. Adipocytes lack antigen-presenting cells, while there are fewer adipocytes in the deltoid muscle, which is conducive to antigen presentation to T or B cells [Lemon, S. M. 1986, J Gen Intern Med]. In addition, studies have shown that intradermal influenza vaccine can produce an immune response that is as effective as intramuscular injection with less amount of immunogen [Arnou R, et al. 2010,Hum Vaccin,2010.]. Therefore, the appropriate route of administration should be determined according to the nature of different vaccine preparations and the purpose of vaccination.”

Comment 5: A table containing the VLP formulations on the market, their composition, the administration route and specific references should be added in order to improve the quality of the manuscript.

Response: As suggested, we provide a table (the new Table 1) showing the VLP formulations on the market or in clinical development, including their composition, the administration route and the corresponding references.

Round 2

Reviewer 1 Report

The revised version of the manuscript addresses most of my previous comments and adds more relevant information about VLPs and their functions. Table 1 was a good addition. As such, most of my comments are addressing minor English or grammatical issues. I do have several minor comments regarding the new information that has been added to the revised version that should be addressed before publication. After these issues are addressed, I believe the manuscript should be accepted.

Minor comments:

- Lines 271-279: The explanation of consensus VLPs is insufficient and the wording is confusing. Please change to, for example "In addition to VLPs containing antigens against a single viral strain, "consensus VLPs" that were designed with multiple antigens from different strains, can be used with the goal to potentially induce cross-protective immunity. The GII.4 consensus norovirus VLP..."

- No human data regarding the norovirus VLP-based candidate has been provided in the text. Currently, a bivalent VLP-based vaccine consisting of the consensus GII.4 VLP and a GI.1 VLP is in Phase II clinical trials [Atmar et al., 2018, Journal of Infectious Diseases]. A short description of this VLP-based vaccine candidate would fit in well here.

- Some basic information is missing from the short description of the consensus COBRA 2 influenza vaccines candidate, such as what COBRA stands for. Please add an additional sentence or two to provide more details about this vaccine candidate.

- VLPs are defined as being free of genetic material, yet in Figure 2, both VLPs have genetic material in them. Please clarify in the text.

- Table 1: The Vaxart norovirus vaccine candidate is not VLP-based. The vaccine candidate consists of a non-replicative, recombinant adenovirus vector that expresses the norovirus VP1 protein. Please remove this from the table.

- Please reference Figure 1 in the text.

- Lines 261-264: To simplify the sentence and help with clarification, please change. For example: "The antigen in the RTS,S/AS01 vaccine consists of a recombinant form of circumsporozoite protein (CSP) linked to HBsAg(S) via an amino acid repeat and T-cell epitope domain. These proteins can spontaneously enter the cells to form VLPs that subsequently serve as a vaccine."

- Table 1: The P2-VP8-P[8] rotavirus vaccine was produced by Walter Reed Army Institute of Research.

- Line 124-125: The sentence beginning with "Covalently combining..." Is confusing. Please reword.

Minor English/Grammar Comments:

- Line 53: Please change "VLP can be divided" to "VLPs can be divided".

- Lines 56-67: Please change to "Enveloped VLPs, such as influenza VLPs, have a complex structure constituted by both virus and host membrane components."

- Line 58: Please change to "In addition to their capability to stimulate an immune response..."

- Line 88: Please change to "which substantially enhance the stimulation of the immune system..."

- Lines 89-92: This sentence is long and contains a lot of information. Please divide into two sentences, if possible.

- Line 97: Please change to "doxorubicin (DOX) mixed with..."

- Line 127: Please change to "...overexpressed on tumor cells. Covalently linking FA..."

- Line 129: Please change to "can achieve cell-specific delivery."

- Line 167: Please change to "Due to..."

- Line 197: Please change to "can self-assemble into VLPs"

- Lines 231-232: Please change to "...is the agent of enterically transmitted hepatitis with a worldwide distribution, and may cause severe acute hepatitis. The genome length of HEV is 7.2 kb; it includes three..."

- Line 254-255: Please change to "...is capable of eliciting antibodies that target..."

Author Response

Summary:

We would like to thank the two reviewers for their constructive comments which greatly improve our manuscript. We have addressed all the comments of the reviewers raised in the second round of review.

============================================================

Reviewer #1

Comments and Suggestions for Authors
Reviewer: The revised version of the manuscript addresses most of my previous comments and adds more relevant information about VLPs and their functions. Table 1 was a good addition. As such, most of my comments are addressing minor English or grammatical issues. I do have several minor comments regarding the new information that has been added to the revised version that should be addressed before publication. After these issues are addressed, I believe the manuscript should be accepted.
Response: We thank the reviewer for reviewing our revision and conveying encouraging comments. We have further revised the manuscript according to the suggestions.

Specific comments

Minor comments:
Comment 1:
Lines 271-279: The explanation of consensus VLPs is insufficient and the wording is confusing. Please change to, for example "In addition to VLPs containing antigens against a single viral strain, "consensus VLPs" that were designed with multiple antigens from different strains, can be used with the goal to potentially induce cross-protective immunity. The GII.4 consensus norovirus VLP..."
Response: As suggested, we have changed the sentence “In addition to the common VLP vaccine form, a new VLP candidate vaccine, named "Consensus VLP", has been used in the development of VLP-based vaccines for noroviruses and influenza viruses.” to “In addition to VLPs containing antigens against a single viral strain, "consensus VLPs" that were designed with multiple antigens from different strains, can be used with the goal to potentially induce cross-protective immunity. The GII.4 consensus norovirus VLP...”

Comment 2: No human data regarding the norovirus VLP-based candidate has been provided in the text. Currently, a bivalent VLP-based vaccine consisting of the consensus GII.4 VLP and a GI.1 VLP is in Phase II clinical trials [Atmar et al., 2018, Journal of Infectious Diseases]. A short description of this VLP-based vaccine candidate would fit in well here.

Response: As suggested, we have added the following information about the consensus norovirus VLP.

“Currently, a bivalent VLP-based vaccine consisting of the consensus GII.4 VLP and a GI.1 VLP is in phase 2 clinical trial. Formulated with the adjuvant containing MPL and aluminum hydroxide and administered intramuscularly, this vaccine showed well tolerated, safety and reactogenicity in the clinical trial [Atmar et al., 2016, Journal of Infectious Diseases].”

Comment 3: Some basic information is missing from the short description of the consensus COBRA 2 influenza vaccines candidate, such as what COBRA stands for. Please add an additional sentence or two to provide more details about this vaccine candidate.

Response: As suggested, we detailed the sentence to “In addition, in order to cope with the diversity of the H5N1 influenza virus, a chimeric hemagglutinin (HA) based vaccines were developed by a methodology termed computationally optimized broadly reactive antigen (COBRA), named Human COBRA 2. The Human COBRA 2 incorporated key neutralizing epitopes of …”

Comment 4: VLPs are defined as being free of genetic material, yet in Figure 2, both VLPs have genetic material in them. Please clarify in the text.

Response: As suggested, we have expanded the notes in Figure 2 as: “The internal cavity of the VLPs accommodates exogenous oligonucleotides (rendered as brown gene-like patches), drug molecules (red circle), and small proteins (green pentagram) that can be delivered after the VLPs get the target organs or cells. The outer surface of the VLPs displays foreign proteins that are fused (red triangle) or covalently conjugated (yellow-green circle) to the particles.”

Comment 5: Table 1: The Vaxart norovirus vaccine candidate is not VLP-based. The vaccine candidate consists of a non-replicative, recombinant adenovirus vector that expresses the norovirus VP1 protein. Please remove this from the table.

Response: As suggested, we have removed the Vaxart norovirus vaccine from table 1.

Comment 6: Please reference Figure 1 in the text.

Response: As suggested, we have referenced Figure 1 in the sentence: “Here, we review the utility of VLPs in drug delivery and antigen presentation, and highlight the recent advances in VLP-based vaccine research (Figure 1)”.

Comment 7: Lines 261-264: To simplify the sentence and help with clarification, please change. For example: "The antigen in the RTS,S/AS01 vaccine consists of a recombinant form of circumsporozoite protein (CSP) linked to HBsAg(S) via an amino acid repeat and T-cell epitope domain. These proteins can spontaneously enter the cells to form VLPs that subsequently serve as a vaccine."

Response: As suggested, we have changed the sentence to "The antigen in the RTS,S/AS01 vaccine consists of a recombinant form of circumsporozoite protein (CSP) linked to HBsAg(S) via an amino acid repeat and T-cell epitope domain. These proteins can spontaneously enter the cells to form VLPs that subsequently serve as a vaccine."

Comment 8: Table 1: The P2-VP8-P[8] rotavirus vaccine was produced by Walter Reed Army Institute of Research.

Response: Thanks for your information, we have changed the “Silver Spring, MD, USA” to “Walter Reed Army Institute of Research “ in Table 1.

Comment 9: Line 124-125: The sentence beginning with "Covalently combining..." Is confusing. Please reword.

Response: We apologize for this confusion. We have changed this sentence to “Some exogenous RNA operons can trigger the assembly of MS2 VLPs to package drugs, then be delivered to specific cells which is mediated by some specific guiding molecules that have been covalently conjugated to VLPs.”

Minor English/Grammar Comments:

Comment 1: Line 53: Please change "VLP can be divided" to "VLPs can be divided".

Response: As suggested, we have changed "VLP can be divided" to "VLPs can be divided"

Comment 2: Lines 56-67: Please change to "Enveloped VLPs, such as influenza VLPs, have a complex structure constituted by both virus and host membrane components."

Response: We have changed the sentence to "Enveloped VLPs, such as influenza VLPs, have a complex structure constituted by both virus and host membrane components."

Comment 3: Line 58: Please change to "In addition to their capability to stimulate an immune response..."

Response: We have changed the sentence to "In addition to their capability to stimulate an immune response...".

Comment 4: Line 88: Please change to "which substantially enhance the stimulation of the immune system..."

Response: We have changed the sentence to "which substantially enhance the stimulation of the immune system...".

Comment 5: Lines 89-92: This sentence is long and contains a lot of information. Please divide into two sentences, if possible.

Response: We have rephrased the sentence in two separate sentences, now reads, “Foreign antigens display on the VLP surface can be achieved by gene engineering or chemical cross-linking. One way is to insert foreign antigens into some specific sites within the viral structural protein by fusion protein expression [21], the other way is to chemically decorate the VLP surface with target antigens through bifunctional cross-linkers, such as 4-(N-maleimidemethyl) loops Hexane-1-carboxylic acid sulfosuccinimide ester (sulfo-SMCC), nanogels and so on [22].”

Comment 6: Line 97: Please change to "doxorubicin (DOX) mixed with..."

Response: We have changed the sentence to "doxorubicin (DOX) mixed with...".

Comment 7: Line 127: Please change to "...overexpressed on tumor cells. Covalently linking FA..."

Response: We have changed the sentence to "...overexpressed on tumor cells. Covalently linking FA...".

Comment 8: Line 129: Please change to "can achieve cell-specific delivery."

Response: We have changed the sentence to "can achieve cell-specific delivery".

Comment 9: Line 167: Please change to "Due to..."

Response: We have changed to "Due to...".

Comment 10: Line 197: Please change to "can self-assemble into VLPs"

Response: We have changed to "can self-assemble into VLPs".

Comment 11: Lines 231-232: Please change to "...is the agent of enterically transmitted hepatitis with a worldwide distribution, and may cause severe acute hepatitis. The genome length of HEV is 7.2 kb; it includes three..."

Response: We have changed the sentence to  "...is the agent of enterically transmitted hepatitis with a worldwide distribution, and may cause severe acute hepatitis. The genome length of HEV is 7.2 kb; it includes three...".

Comment 12: Line 254-255: Please change to "...is capable of eliciting antibodies that target..."

Response: We have changed the sentence to "...is capable of eliciting antibodies that target...".

Reviewer 2 Report

The authors performed the required revision. The manuscript can be accepted in current form.

Author Response

Thank you for your confirmation on our response. We indeed appreciate your comments in the first round review, which greatly help to improve our manuscript.